# The transmission of pottery technology among prehistoric European hunter-gatherers

Ekaterina Dolbunova [1,2,30], Alexandre Lucquin [3,30],
T. Rowan McLaughlin [2,28,30] ✉, Manon Bondetti[3], Blandine Courel[2],
Ester Oras [4], Henny Piezonka[5], Harry K. Robson [3], Helen Talbot[3],
Kamil Adamczak [6], Konstantin Andreev[7], Vitali Asheichyk [8,29],
Maxim Charniauski [8], Agnieszka Czekaj-Zastawny [9], Igor Ezepenko[8],
Tatjana Grechkina[10], Alise Gunnarssone[11], Tatyana M. Gusentsova[12],
Dmytro Haskevych [13], Marina Ivanischeva[14], Jacek Kabaciński [9],
Viktor Karmanov [15], Natalia Kosorukova[16], Elena Kostyleva[17], Aivar Kriiska [4],
Stanisław Kukawka[6], Olga Lozovskaya [18], Andrey Mazurkevich[1],
Nadezhda Nedomolkina[19], Gytis Piličiauskas[20], Galina Sinitsyna[18],
Andrey Skorobogatov [21], Roman V. Smolyaninov [22], Aleksey Surkov[23],
Oleg Tkachov [8], Maryia Tkachova [8], Andrey Tsybrij[24], Viktor Tsybrij[24],
Aleksandr A. Vybornov [7], Adam Wawrusiewicz[25], Aleksandr I. Yudin [26],
John Meadows [27], Carl Heron [2] & Oliver E. Craig [3]

Human history has been shaped by global dispersals of technologies, although understanding of what enabled these processes is limited. Here, we explore the behavioural mechanisms that led to the emergence of pottery among hunter-gatherer communities in Europe during the mid-Holocene. Through radiocarbon dating, we propose this dispersal occurred at a far faster rate than previously thought. Chemical characterization of organic residues shows that European hunter-gatherer pottery had a function structured around regional culinary practices rather than environmental factors. Analysis of the forms, decoration and technological choices suggests that knowledge of pottery spread through a process of cultural transmission. We demonstrate a correlation between the physical properties of pots and how they were used, reflecting social traditions inherited by successive generations of hunter-gatherers. Taken together the evidence supports kinship-driven, super-regional communication networks that existed long before other major innovations such as agriculture, writing, urbanism or metallurgy.

The dispersal of new technologies is central to the evolution of cultural systems globally. Analysis of archaeological materials to track the rate and direction that ancestral technologies spread, and the behavioural mechanisms that led to their adoption, are important enquiries in the study of cultural evolution. A major advance has been to track the spread of farming and associated technologies during the Early Holocene, using large repositories of radiocarbon-dated cultural material[1]. It has been shown that in most parts of Europe, the

**Fig. 1 | Study area, site locations and examples of reconstructed forms for the pottery styles included in this study.** Illustrated are reconstructions from the (1) Eastern Baltic, (2) Western Baltic, (3) Upper Dnieper, (4) Bug-Dniester, (5) Middle Don, (6) Lower Don, (7) Northern Caspian, (8) Lower Volga, (9) Middle Volga and (10) Upper Volga regions. Map based on the ASTER Global DEM v.3 with ecotones based on generalized mid-Holocene estimates from ref. [91]; it should be noted that the boundary between steppe and forest is likely to have been highly diffuse.

process is satisfactorily explained through demic diffusion[2–6], in which an expanding population carries with it a coherent package of technologies associated with domesticated plants and animals. Here, innovations arise relatively slowly, resulting in a recognizable 'package' that is maintained across the dispersal trajectory. Hunter-gatherer societies have a subsistence base involving hunting, foraging and fishing with little reliance on domesticates. Compared with farming societies, the innovation and transmission of other fundamental technologies by prehistoric Holocene hunter-gatherers is not well understood, partly because there are fewer opportunities for obtaining behavioural parallels from contemporary communities, especially those from comparable temperate environments, and partly because of a much sparser archaeological record. Yet such studies are vital if we are to appreciate the role of ancestral hunter-gatherers in shaping cultural and social systems.

Here we report an important advancement of knowledge regarding the dispersal of pottery containers; a hunter-gatherer innovation that spread to become ubiquitous globally. Pottery first emerged among East Asian hunter-gatherers towards the end of the Late Pleistocene[7,8]. Regression models based on radiocarbon dates of the arrival times suggest that pottery spread from East Asia across Northern Eurasia during the Early Holocene[9]. Yet, this analysis on a pan-continental scale fails to elucidate the mode of transmission, nor is it able to rule out multiple independent innovations in pottery, or address what the functional needs for pottery by diverse hunter-gatherers might have been. Likewise, previous super-regional analysis of hunter-gatherer pottery transmission[10] is founded on radiocarbon chronologies complicated by the varying reliability of the materials and contexts dated[11]. Overall, our understanding of how, why and when this phenomenon dispersed is inadequate.

Focusing on the vast East European plain (Fig. 1), a key potential conduit for the westward dispersal of pottery by hunter-gatherers

during the sixth millennium BC, we aim to test three related hypotheses. First, that the dispersal process was continuous rather than derived from multiple origins. Second, that demic processes of population expansion led to the spread of pottery. Third, that the process was driven by an underlying socio-economic need resulting in functional similarity across the study region. With no existing dataset to draw upon, we tested these hypotheses by directly analysing pottery from 156 European hunter-gatherer sites (Fig. 1) to generate models of cultural transmission using primary data gathered from 1,491 potsherds from 1,226 vessels and the associated radiocarbon dates. Without major mountain ranges, the study area is highly conducive to human mobility, with only forested morainic hills and upland areas in the Don or Volga catchments as potential impediments. The majority of sites are settlements represented by various pits, platforms, artefact scatters and other ephemeral structures, often located close to major rivers or their tributaries[12]. Faunal and botanical analyses have shown that a broad spectrum of hunted, gathered and fished resources was exploited across the study area[13–15].

Attributes related to production, such as shape, size, decoration and method of manufacture were obtained from a representative sample of pottery from each site (Methods: Sampling rationale and Ceramic data acquisition). These attributes, sometimes taken together and interpreted as 'archaeological cultures', represent human knowledge fossilized in the artefact. They can be used to reconstruct connections between societies, separated by geographic distance or time, using a set of biostatistical tools to evaluate the relatedness of archaeological cultures based on traits[16,17]. Functional attributes related to use were obtained through lipid residue analysis of the vessels, using standardized methodologies[11]. We present here an amalgamated dataset of new residue analyses of 552 pottery vessels or adhering charred surface deposits (foodcrusts), and revised data from 674 vessels previously published from across the study region (Supplementary Table 1).

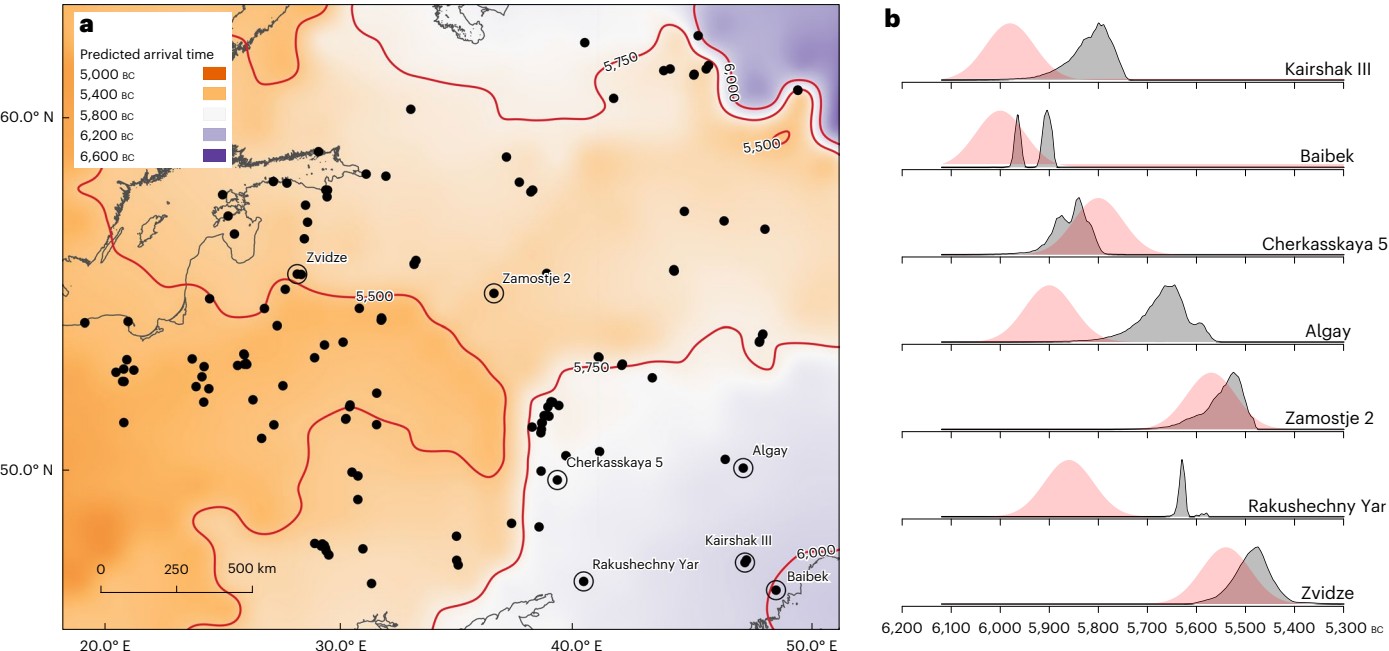

**Fig. 2 | A spatiotemporal model for the spread of pottery technology. a**, Predicted start date for pottery vessels in the region based on spatial–temporal regression models. **b**, The modelled posterior probability distribution for selected locations based on radiocarbon evidence compared to the predicted dates. For details on how the models are constructed, see Supplementary Methods.

The samples chosen for lipid residue analysis were quantitatively representative of the broader assemblage in terms of morphological, stylistic and technical attributes.

## Results

### Dating and spatial–temporal modelling

New radiocarbon dates and age models reveal that pottery appeared near the northern shore of the Caspian Sea shortly before 5900 cal BC, and spread rapidly northwards and westwards (Supplementary Methods: Site chronologies). However, direct radiocarbon dating of pottery is complicated because of the ubiquity of freshwater- and marine-derived carbon present in foodcrusts that tend to produce dates significantly older than the use of the vessel[18]. To circumvent these 'reservoir effects', likely arrival dates for pottery were built for selected sites using multiple terrestrial samples of bone and charcoal found in direct association (Fig. 2). Although isolated cases of innovation cannot be excluded, regression models[2] extrapolated over the study area based on these dates are consistent with a continuous process of adoption with the earlier occurrence of an antecedent tradition in western Siberia or central Asia (Methods: Spatial–temporal modelling). An origin in western Siberia provided a better fit for the data than central Asia (Fig. 2a), although the predicted arrival times based on both points of origin are not significantly different from each other and are consistent with an ultimate origin for these traditions in the Far East[19]. Crucially, the regression models suggest an average rate of diffusion of 6–10 km yr$^{-1}$, several times faster than, for example, the spread of farming in Western Europe[20,21], representing accelerated expansion across the study area compared with the Eurasian average of 0.2–1.2 km yr$^{-1}$ (ref. [19]). At certain sites, notably Rakushechny Yar in the Lower Don, radiocarbon evidence shows that the sampled ceramics derive from occupation several centuries later than when the regression models suggest pottery first appeared in the locality. In other cases, such as the Zedmar culture pottery of the Prussian lowlands and Masurian Lake District, much later dates are reported[22]. These ceramics are unlikely to be part of the initial dispersal of hunter-gatherer ceramics and are excluded from the statistical analysis of stylistic and technological traits

because they are the product of later phenomena and influences from multiple sources, including agricultural societies[22].

### Organic residue analysis of vessel use

Lipid residue data are reported for the entire sample set (1,491 samples from 1,226 vessels) of hunter-gatherer pottery from the sites detailed in Supplementary Table 1. Using the acidified methanol extraction procedure[11], >95% (n = 1,425) of the samples yielded lipid quantities above the threshold amount required for interpretation (>5 μg g$^{-1}$ for potsherds and >100 μg g$^{-1}$ for charred surface deposits) or contained distinctive lipids traceable to a specific source. In addition, 100 samples were also solvent-extracted following established procedures[11] to investigate either the presence and distribution of triacylglycerols or the presence of other intact lipids (for example, wax esters). These failed to provide additional information. We assigned the residues to different classes of product (aquatic fats, ruminant animal fats and plant oils) based on multiple molecular and isotopic criteria (Methods) by gas chromatography, gas chromatography–mass spectrometry (GC–MS) and gas chromatography–combustion–isotope ratio mass spectrometery (GC–C–IRMS). In cases in which multiple products were attributable to a single vessel (for example, aquatic lipids, ruminant fats) each product was included in the overall count. Residues absorbed within the vessel wall and those obtained from charred deposits on the same vessel were treated as separate cases. This count has to be considered as a minimal conservative number of occurrences of a resource because the absence of certain criteria is not always related to the absence of a resource.

Fatty acid stable isotope data obtained by GC–C–IRMS of 1,272 samples of hunter-gatherer pottery from all phases are plotted in Fig. 3. About half of the samples analysed yielded lipid biomarkers typical of aquatic organisms (709 of 1,425) and these tended to have a broader range of $\delta^{13}C_{16:0}$ and $\delta^{13}C_{18:0}$ values representing extreme freshwater and marine carbon isotopic end-points (Fig. 3a). Vessels without aquatic biomarkers have a narrower distribution of $\delta^{13}C_{16:0}$ and $\delta^{13}C_{18:0}$ values (Fig. 3b) and generally more negative $\Delta^{13}C$ values ($\delta^{13}C_{18:0} - \delta^{13}C_{16:0}$) reflecting the input of a higher proportion of

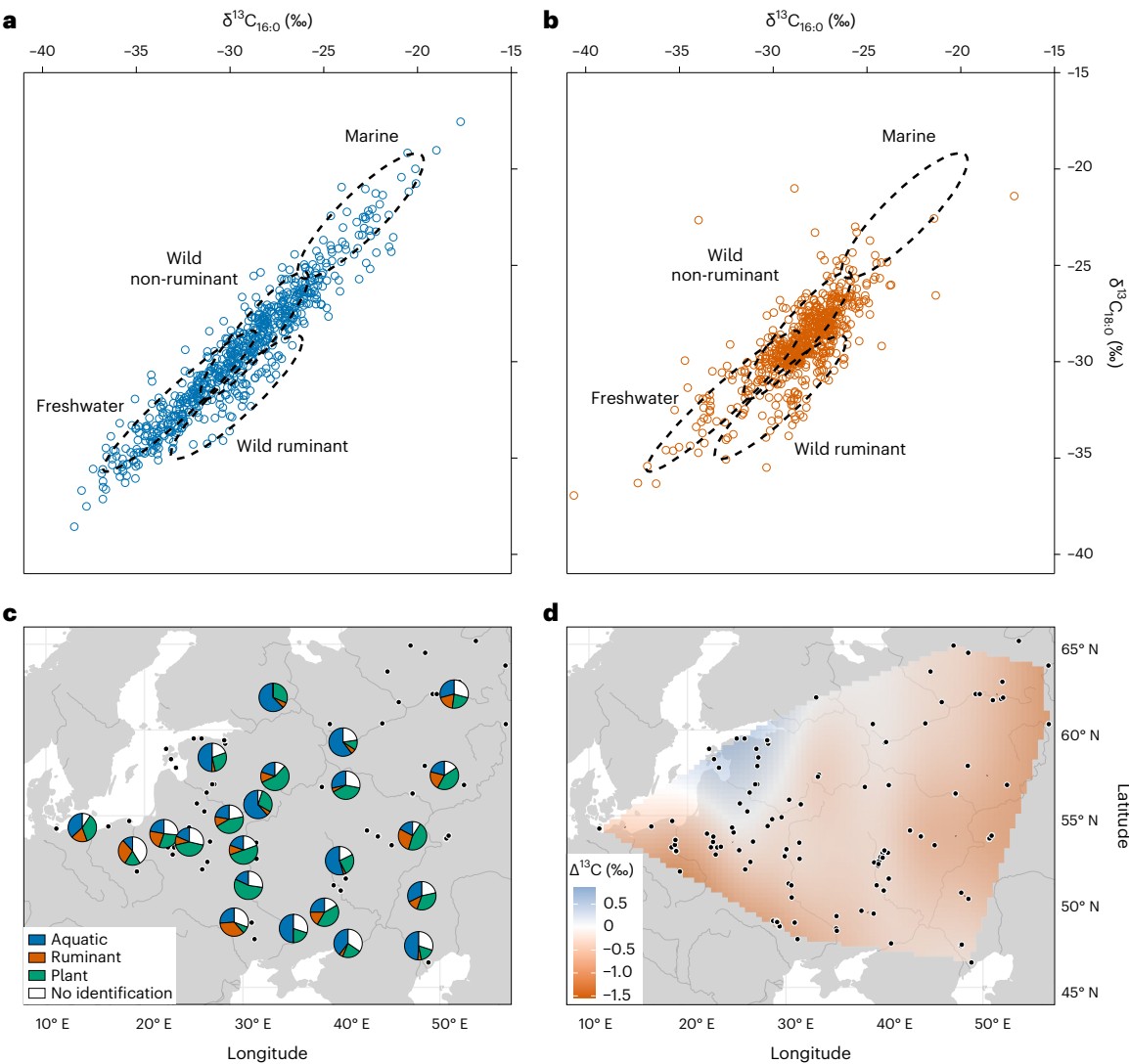

**Fig. 3 | Lipid residue analysis of hunter-gatherer ceramics. a,b,** $\delta^{13}C_{16:0}$ and $\delta^{13}C_{18:0}$ values of sample with (**a**) and without (**b**) aquatic biomarkers and statistical ellipses (1σ) of modern animal fats from Eastern Europe (provided online; see Data Availability Statement). **c,** Relative frequency of identified food commodities by region following the criteria described in the Methods. **d,** A surface model interpolating $\Delta^{13}C$ values (for error maps and other results, see Supplementary Fig. 12).

ruminant fats—presumably wild ruminants such as deer[23]. Despite these broad trends, the isotope values do not cluster within the ranges expected for authentic foodstuffs, pointing to mixing of contents either in single episodes or, perhaps more likely, throughout the life history of the container. Over half of the samples that yielded lipids (814 of 1,425) showed molecular evidence of thermal alteration, which together with the frequent occurrence of carbonized deposits, suggests cooking rather than storage. Plant products are frequent (587 of 1,425), sometimes with fragments of carbonized plant tissues visible within the charred deposit[12,24], but were probably not the main commodities. Typical clear leafy plant lipid profiles are rare and plant biomarkers are generally identified in only small or trace quantities. In 74% of their instances they are associated with aquatic or terrestrial animal fats. There is an almost complete absence (29 of 1,425) of lipid profiles typical of plant resins and tars (where di- or triterpenes are prominent in the extract), perhaps unexpected given the presumed importance of these substances to hunter-gatherers[25,26]. Similarly, only one sample found at Grube-Rosenhof LA 58 (ref. 11) contained beeswax, contrasting with a much higher prevalence in Early Neolithic agricultural pottery[27]. The absence of beeswax was noted even in temperate

regions where honey bees would be expected to thrive. Overall, the residue data overwhelmingly show that hunter-gatherer pottery was primarily a culinary technology.

These data are further disaggregated by region in Fig. 3c and show sub-regional variation in pottery use as noted in previous studies[11] despite broad similarities in environmental settings and resource availability. Generally, aquatic products dominate in the southeastern and central part of the study area, whereas ruminant products were processed more prominently in pottery from the west and northeast. This is also supported by interpolating the $\Delta^{13}C$ values spatially with more negative values corresponding to ruminant products (Fig. 3d). As previously suggested[11], it is likely that such sub-regional 'cuisines' arose due to local customs of food preparation and consumption, and that certain sites were highly specialized[28].

## Analysis of pottery production traits

A set of contingency tables recording the presence or absence of 'production traits' was generated and the relationship between sites was examined using correspondence analysis[29]. Sub-regional styles of pottery production could be identified that roughly recapitulate

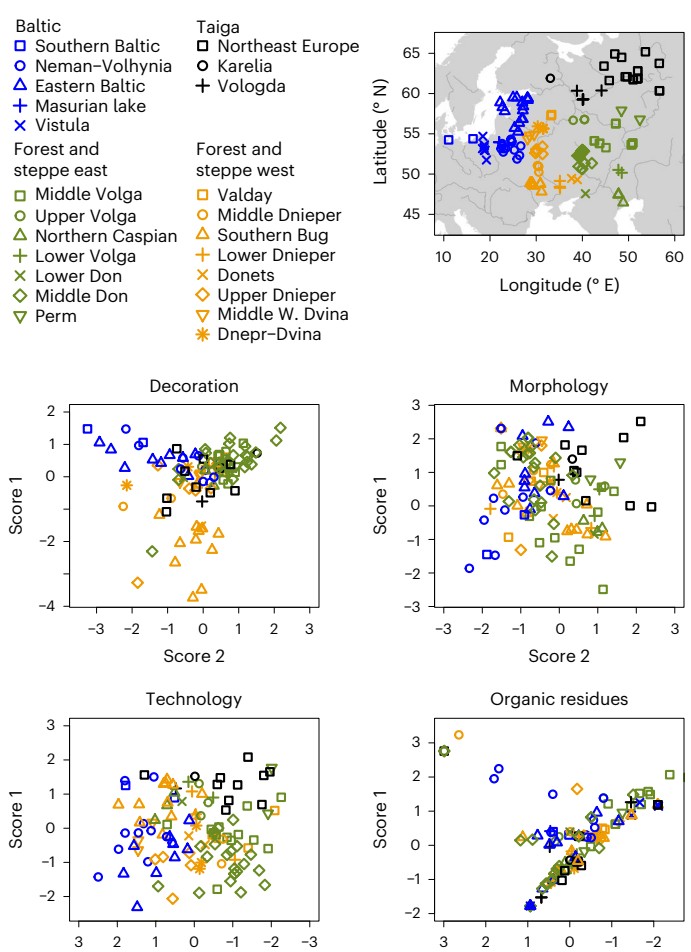

**Fig. 4 | Biplots of correspondence analysis scores for traits (site-wise) recorded through inspection of the archaeological ceramics and the organic residue analysis.** Axes orientation chosen to best illustrate the recapitulation of geographic coordinates.

geography and the major river basins (Fig. 4); this is seen most clearly in the technological traits, providing empirical evidence that technological traditions are embedded within local cultural practices[30,31]. A far weaker pattern was observed when 'use traits' are subjected to correspondence analysis (Fig. 4).

Next, we computed Mantel coefficients to examine the degree of correlation between pottery production and use. Mantel tests were used to compare two distance matrices: spatial distance was determined from site location data, whereas temporal distance was derived from radiocarbon chronologies or inferred from regression models (Methods: Spatial–temporal modelling). Cultural distance, including the 'distance' between the biomolecular traits of pairs of sites, was enumerated using the Jaccard dissimilarity index. Jaccard distances were lower on average for traits associated with vessel use compared with those associated with pottery production, although their variability is higher (Supplementary Table 8). Thus despite regional variation, use of the pots was more consistent over the study region than the cultural factors that influenced the way in which they were made. This is most likely because pottery use was constrained by the relatively homogenous ecological settings, where wild aquatic and forest species were abundant. Inter-site differences in pottery use are not, therefore, caused by gradual processes of geographical isolation, explaining the lack of clear geographic patterning in the correspondence analyses scores.

A robust set of correlations was observed between technology, morphology, decoration and their functional criteria (organic residues) using the Mantel test (Fig. 5), reported here as correlation coefficients ($r$) and associated $P$ values (two-tailed, null hypothesis $r = 0$). Correlations were observed in the three separate domains of ceramic morphology ($r = 0.13$, 95% confidence interval (CI) 0.1 to 0.16, $P \approx 0.001$), technology ($r = 0.18$, 95% CI 0.15 to 0.22, $P \approx 0.001$) and decoration ($r = 0.14$, 95% CI 0.11, $P \approx 0.001$ to 0.17). Combined in a contingency table containing all 129 traits, and using a partial Mantel test to hold geographic distance constant while regressing the Jaccard distance matrices of all pottery traits and organic residues, the correlation coefficient ($r$) is 0.22 (95% CI 0.18 to 0.25, $P \approx 0.001$). As expected, inter-site distance was correlated with pottery traits in terms of technology ($r = 0.25$, 95% CI 0.22 to 0.28, $P \approx 0.001$), but more weakly with morphology ($r = 0.17$, 95% CI 0.15 to 0.20, $P \approx 0.001$) and decoration ($r = 0.12$, 95% CI 0.09 to 0.14, $P \approx 0.002$), and, importantly, was not correlated with organic residue usage traits ($r = 0.02$, 95% CI −0.01 to 0.04, $P \approx 0.77$). Spatio-temporal distance does not correlate with any of the traits, ruling out any pattern of convergent or parallel evolution occurring between contemporary sites separated by large tracts of geographical distance (Supplementary Table 7). Similarly, when we examined Jaccard matrices between pottery technology, morphology, decoration and use traits at the level of the vessel, rather than site, they remained positively correlated despite a loss of statistical power due to the highly fragmented nature of the assemblages (Supplementary Table 9). Overall, there is considerable congruence in the transmission of knowledge regarding hunter-gatherer pottery production and function. These observations also hold for regional subsets of the data (Supplementary Table 10), and when our sample is stratified by vegetation zones (Supplementary Table 11).

Next, we determined the geographic scale over which coherent patterns in the trait data appear by computing Mantel correlograms[32]. These identify spatial autocorrelations in the traits between each site and all other sites in various sets of expanding geographic distances (Fig. 6). Significant positive correlations were observed for pottery morphology (at 100 km, $r = 0.12$, 95% CI 0.10 to 0.15, $P \approx 0.001$) decoration (at 100 km $r = 0.13$, 95% CI 0.11 to 0.16, $P \approx 0.001$) and technology (at 100 km $r = 0.16$, 95% CI 0.14 to 0.19, $P \approx 0.001$), remaining significantly positive within 250–500 km of each site. Significantly negative correlations exist beyond 500–700 km (for example, decoration at 1,000 km, $r = −0.07$, 95% CI −0.09 to −0.05, $P \approx 0.002$).

This provides an insight into the distances over which knowledge of pottery production was directly transferred between prehistoric hunter-gatherer societies, occurring, for example, through direct contact, migrations or marriage networks. Again, no geographic pattern is present in the organic residue data, principally because of similarities in subsistence practices throughout the region.

At a larger scale, we have been able to recover correlations between vessel technology, morphology, decoration and use that are not due to spatial autocorrelation. This discovery, a case of 'form following function', hints at a deeper symbolism employed by the makers of the pots and communicated via some mechanism of cultural transmission throughout the communities involved. To further develop this idea, we modelled the trait data as neighborNets to investigate whether the data are best characterized by a model of branching-and-blending rather than a simple branching phylogeny. The results (Fig. 7) indicate a strong level of input from blending processes, supporting the dominance of cultural transmission as the mechanism behind the spread of pottery[33]. The sites are modelled neighbouring other sites located nearby in either time, space or both, with no two sites modelled in the same clade.

## Discussion

Understanding the mode and tempo of hunter-gatherer pottery dispersal into the European continent sheds light on the mechanisms responsible for cultural transmission in this context. The patterns in

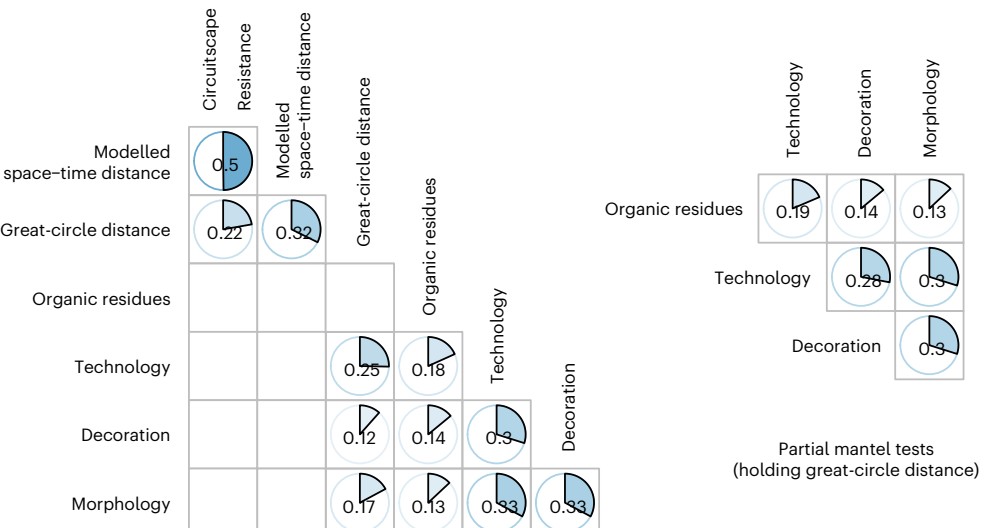

**Fig. 5 | Correlation coefficients for pairwise Mantel tests for Jaccard and geographical distance matrices.** The partial Mantel tests indicate the strength of the correlation between organic residues and pottery characteristics when holding a great-circle distance constant. The Mantel correlation of distance matrices tests a null hypothesis that there is no relationship between the cultural, biomolecular and geographical 'distance'. For cases in which the null hypothesis was rejected, the Pearson correlation coefficients $r$ produced by these Mantel tests are illustrated, with $P$ values (two-sided, null hypothesis $r = 0$) and 95% CIs contained in Supplementary Table 7.

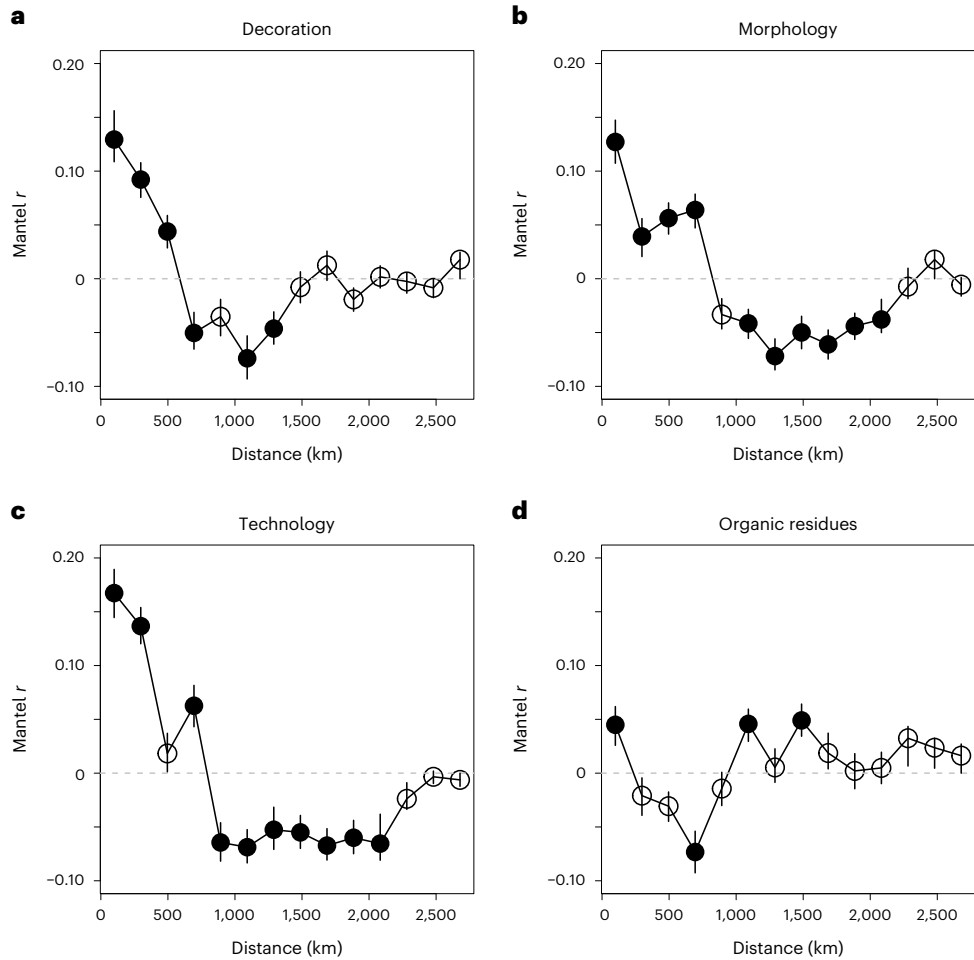

**Fig. 6 | Mantel correlograms showing the scale of spatial autocorrelation.** Significant similarity (Mantel $r > 0$) or dissimilarity (Mantel $r < 0$) is indicated by filled circles. Error bars indicate bootstrapped 95% CIs.

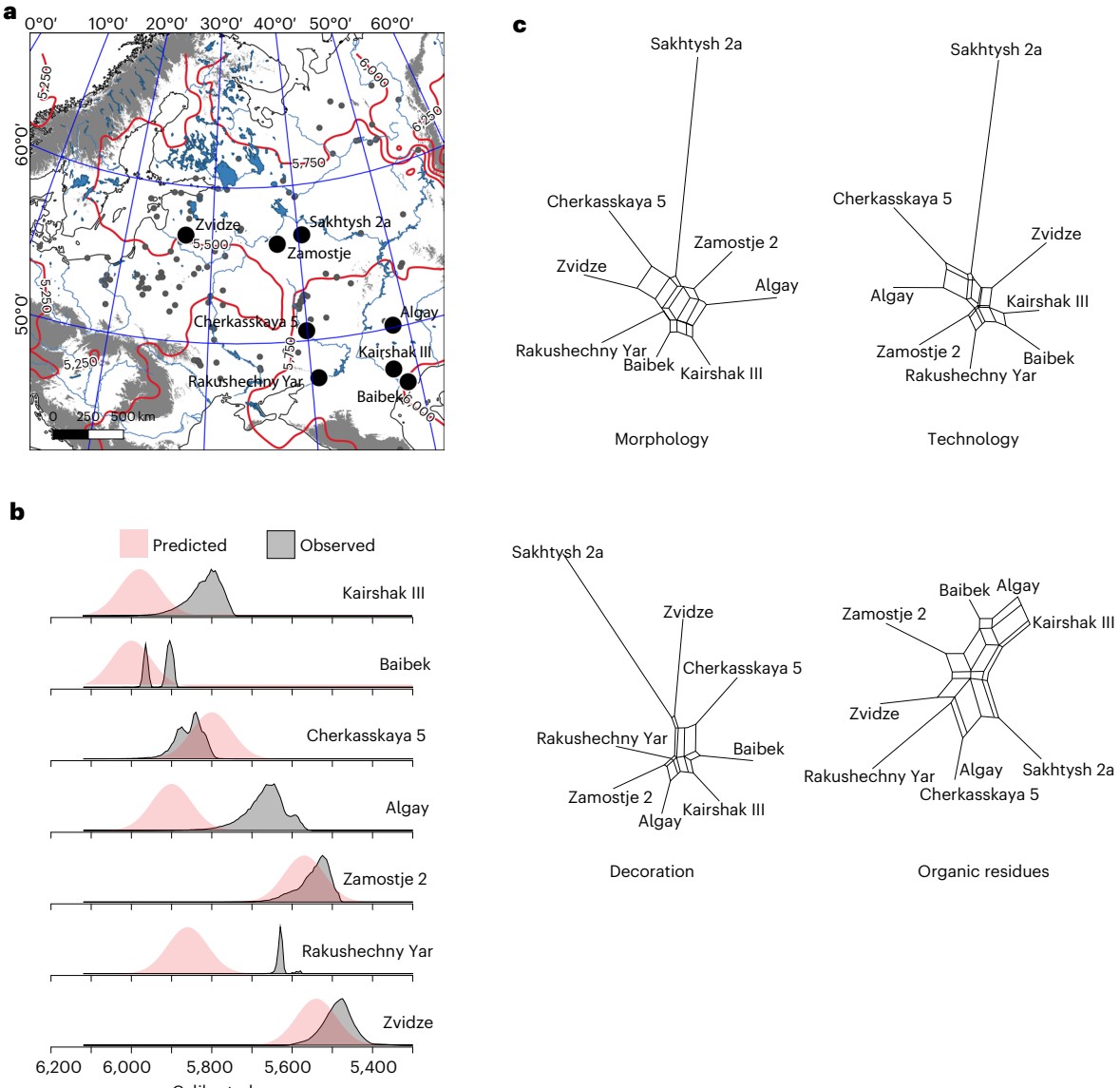

**Fig. 7 | The spatial, temporal and network proximity of selected sites in the study. a**, Locations for sites with isochrones representing a spatio-temporal model of diffusion. **b**, Posterior probability distributions of the date of the start of pottery use at each site. **c**, NeighborNet networks for ceramic and biomolecular dissimilarity data.

our data exist despite several limitations, especially the many factors that dictate what survives in the archaeological record. For example, our data derive from palimpsests that do not necessarily represent the very earliest phase of pottery use in each locality, thereby introducing noise into the spatio-temporal regression models and reducing our capacity to recapture nuances of the behaviour associated with transmission of this technology. Furthermore, lipid residue analysis is strongly biased towards the identification of lipid-rich animal tissues and the approach might not quantitatively capture the complete range of foodstuffs processed in each vessel, and as such represent a narrow range of foodstuffs available. This potentially has led us to underestimate the real strength of the association between pottery production and usage traits.

The earliest dates for pottery in the study area were obtained north of the Caspian Sea at the site of Baibek, ~5900 cal BC. However, based on our least-cost model it is also conceivable that there was considerable trans-Ural transmission of pottery knowledge, which would support dates of ~5750 cal BC obtained from Kama culture pottery from Pezmog

IV in the most northerly part of Eastern Europe[34]. Pottery subsequently spread rapidly westwards towards the Baltic, covering over 3,000 km in three to four centuries. Notably, this is several times faster than the spread of Neolithic pottery from the Middle East into the Mediterranean and western Europe[19,20,35]. Through forward modelling, it has been shown that demic diffusion can drive the spread of ancient technology in cases in which the rate of spread is much less than what we have determined for hunter-gatherer pottery in Europe[3,36] (Supplementary Table 5). Although demic diffusion may have a role, on the basis of its speed we argue that pottery production was rapidly disseminated through knowledge transfer across established networks between dispersed hunter-gatherer communities[37]. To the west, although not considered by our models, hunter-gatherer interactions with early farming populations could have resulted in influences manifesting in certain shared ceramic traits[38]. Taken together, the transmission of pottery among European hunter-gatherers was one end result of a complex series of wide-ranging social interactions. Compared with later developments like metallurgy, pottery is a relatively low-cost

technology; the required raw materials were abundant, and the knowledge and motor skills needed could have been acquired as part of communally shared behaviours situated within the household or close kinship group[39].

From our food residue results, it appears the demand for pottery was not in response to any specific economic requirement; a wide range of aquatic and terrestrial species were identified with no obvious relationship to the ecological setting, and all of which were exploited well before the arrival of pottery[40]. Although ceramics must have had clear advantages over organic containers for the heat processing of foods, our previous hypothesis that it was adopted in response to more intensive fishing, based on observations in the Eastern Baltic[41], is no longer supported when considering data across the entire study area. Whereas the dispersal of technologies inextricably linked to farming required specific environmental conditions suitable for crop cultivation and the rearing of livestock resulting in marked 'slowdowns'[42], without such constraints, pottery and potentially other hunter-gatherer technologies dispersed much more rapidly. In particular, the mid-Holocene resource-rich forest, coastal, riverine and lacustrine ecotones of Northern Eurasia were an obvious dissemination route akin to other resource-rich 'highways' used to explain the dispersal of hunter-gatherer populations[43], even if the northern forest and taiga environments were less conducive to rapid movement outside river systems compared with open steppe.

Many of the production traits must have had little selective advantage and variability can be largely explained by isolation-by-distance, where innovations occurred gradually due to random copying effects[44]. Conversely, usage traits were, necessarily, more tightly constrained by the relatively homogenous foodscape, but it is nonetheless remarkable that knowledge of pottery techno-function is also transmitted along with decoration, technology and morphology. At its most granular level, this relationship is an example of the mechanism of coherence in social evolution[45], in which traits in different 'things' evolve together because they both reflect deeply rooted social traditions and structured, communal activities. Because culinary practices are often highly structured[46,47], with specific foodstuffs associated with distinct cooking and serving wares, it is no surprise that production- and use-related traits propagate together as a coherent tradition. It is, however, noteworthy that this phenomenon produces a signal that can overcome the appreciable filter imposed by the limited range of foods identifiable using lipid residue analysis. The food residue data are representative of culinary traditions that pass from one community to the next, opening a useful behavioural perspective on the interpretation of datasets traditionally used to reconstruct subsistence practices[48,49].

More broadly, innovation and hybridization, which tend to be accelerated by horizontal transmission[50], must have occurred at a relatively slow pace, or perhaps more likely in sporadic episodes that are difficult to resolve at the scale of our study, otherwise the patterns and groupings we detect in pottery morphology and decoration, sometimes identified as archaeological 'cultures', would not exist. It is an open question how far these 'cultures' can reflect discrete groups of people, wider communication networks or are, in some cases, merely the product of discontinuous sampling from continuous variation[17]. Here, it seems the latter is more conceptually applicable, but that innovation occurred more slowly than adoption through communication networks. Together, these long-recognized evolutionary processes result in delineated and recognizable cultural groups that have shaped the discipline of prehistoric archaeology over much of the twentieth century[51,52].

Our data suggest close technological and stylistic connections between communities located ~250 km apart. Given our estimated dispersal rate 6–10 km yr$^{-1}$, this is consistent with connections encompassing a single human generation (20–30 yr). Genomic analyses of an albeit limited number of human remains from western parts of the study area provide low relative mobility estimates compared with other prehistoric European populations[53]. This may have imposed constraints on how far material culture was spread by any one generation and could explain why the geographic signals in our data only manifest over relatively local scales.

Conversely, our results also bear signals of cultural and economic connectivity that occur throughout the region. The correlations between pottery technology, morphology, decoration and culinary use indicate there were behaviours and symbolic ideas shared by groups located far apart in time and space. Although the idea that hunter-gatherer pottery can spread without significant population movements has been stated before[54,55], a behavioural explanation is still required that can accommodate the loss of cultural traits at relatively local scales, and also the emergence of coherent patterns at much larger scales. Sex-specific demographic behaviour provides one such explanation; for example, the dissemination of female crafts embedded in a patrilocal kinship system, as documented in American Pacific Northwest societies[45]. A similar interpretation has been proposed to explain regional patterning in the later, Corded Ware pottery of the Eastern Baltic[56]. Alternatively, there may have been an element of long-distance exchange or contact. Forager mobility is generally a function of seasonality, subsistence, sources of raw materials and exchange networks. Multiscalar, 'superdiffusive' movements are a fundamental feature of hunter-gatherer landscape use[57,58] and it is thus likely that a combination of mechanisms were at work, including long-distance exchanges. It remains that culinary traditions reflect how technological forms of knowledge were shared among prehistoric hunter-gatherers in Europe. Food was a core element of these cultures, and their pottery represents multiple instances where similar ideas were shared across networks encompassing vast areas.

## Methods

### Sampling rationale
Our study targeted known archaeological assemblages of early hunter-gatherer cooking vessels. Sampling permission was obtained from the site excavation directors and archive holders. The size and composition of each pottery assemblage varied considerably but in all cases potsherds were chosen to maximize the typological variability present at each site. Because there are low occurrences of pottery vessels from some individual sites, our analytical approaches employ resampling procedures to test a null hypothesis that patterns in the traits shared between sites occur randomly. Data collection and analysis were not performed blind to the conditions of the experiments.

### Ceramic data acquisition
A set of presence–absence tables was used to record the pottery features that contain information about the steps of production and use of the vessels: that is, the type of temper and paste, ways of modelling, surface treatment, wall thickness and so on. Together these form the chaînes opératoires, or the sets of social and cognitive acts that are associated with the manufacture of pottery[59–62], although because the assemblage is rather fragmented the whole chaîne opératoire cannot be reconstructed in many cases. Morphometric analyses such as the shape and size classes of the vessels were based on three-dimensional reconstruction, vessel volume calculation and the similarity of vessel profiles and their proportions. A total of 162 traits were recorded: 61 for pottery decoration, 61 for morphology and 40 for pottery technology, such as the type of raw material used for fabric and temper, and vessel modelling and finishing technique. These are described in Supplementary Table 6, and the contingency tables are provided online.

### Typological distance
Using the contingency tables for traits described above, computer scripts in R aggregated these data at the site level. Jaccard dissimilarity indices were calculated for each pair of vessels and sites, using the R package vegan[63] with the results stored in a distance matrix.

## Lipid extraction and analysis

The analytical procedure for lipid extraction followed detailed published methods[11]. Briefly, samples were extracted and methylated in one-step with acidified methanol ($H_2SO_4$/MeOH, 1:5). Methanol was added to homogenized carbonized residues (10–20 mg) or drilled/crushed ceramic powders (0.5–1.0 g), sonicated for 15 min, acidified with concentrated sulfuric acid and the acidified suspension was then heated for 4 h at 70 °C. Lipids were extracted by phase separation with $n$-hexane (3 × 2 ml). Extracts were analysed by GC–MS in total ion current mode for general screening purposes, in selected ion monitoring mode to target specific markers of aquatic resources and by GC–C–IRMS to obtain the carbon isotope values of the most abundant fatty acids ($C_{16:0}$ and $C_{18:0}$). A selection of samples (Supplementary Information) was subjected to solvent extraction[11]. Lipids from ceramic powder were extracted using dichloromethane:MeOH (2:1, 3 × 4 ml), then dried under $N_2$. The extract was trimethyl-silylated using N,O-bis(trimethylsilyl)trifluoroacetamide with 1% trimethylchlorosilane before high temperature GC–MS to detect either the presence and distribution of triacylglycerols or the presence of other intact lipids (for example, wax esters).

The identification of compounds was conducted with Agilent Chemstation and Mass Hunter (Agilent Technologies) software according to their mass spectrum, their retention time and with the help of NIST MS search and NIST 2014 library of mass spectra. Computations of GC–C–IRMS data were made with Isodat (Thermo Fisher) and IonOS software (Elementar).

## Biomolecular criteria for defining organic residue traits

The analytical procedure deployed is suitable for identifying fats, oils and waxes from a wide range of plant and animal products. Using the GC–MS and GC–C–IRMS data, the presence or absence of a range of different food contents (aquatic resources, ruminants, animals and plants) and their processing (heating) was determined for each sample. The 17 interpretative criteria used are detailed below.

(1) (Aquatic) The presence of aquatic-derived lipids (fish, shellfish, aquatic mammals and birds) is inferred from the presence of $\omega$-($o$-alkylphenyl) alkanoic acids (APAAs) with $C_{18}$ and at least $C_{20}$ carbon atoms and isoprenoid fatty acids (either phytanic, pristanic or 4,8,12-trimethyl tridecanoic acid)[64,65].

(2) (Aquatic) $C_{18}$ and $C_{20}$ APAAs can also be derived from terrestrial animal fats. Further refinement of the former criteria can be achieved using the $C_{20}$:$C_{18}$ APAA ratio. Ratios above the tentative threshold of 0.06 are considered to derive from an aquatic source[66].

(3) (Aquatic) The major source of phytanic acid in food-derived fats are aquatic oils and ruminant fats. They can be distinguished by examining the ratio of the two naturally occurring configurations, or diastereomers, of phytanic acid ($3S,7R,11R,15$-phytanic acid (SRR) and $3R,7R,11R,15$-phytanic acid)[67,68]. Despite considerable overlap, the SRR isomer tends to dominate in aquatic oils compared with ruminant fats and a SRR percentage >75.5% can be assigned to this source, using a conservative limit (95% confidence).

(4) (Ruminant) The discrimination of ruminant-derived lipids is generally based on differences in the biosynthesis of fatty acids compared with non-ruminant tissues leading to a depletion in $^{13}C$ of $C_{18:0}$ relative to $C_{16:0}$[69,70]. In reference material from the study area (Supplementary Information), the mean offset $\Delta^{13}C$ ($C_{18:0} - C_{16:0}$) measured in wild ruminant (red deer, roe deer, elk, reindeer and saiga) adipose fats is $-2.28$‰ ± 1.02‰ ($n = 39$), whereas in non-ruminant fats (freshwater and wild non-ruminant terrestrial animals) it is 0.36‰ ± 1.04‰ ($n = 345$), showing a partial overlap of values. Samples with a $\Delta^{13}C$ value

less than $-1.72$‰ (2 s.d. from the non-ruminant mean) have been interpreted as containing ruminant lipids.

(5) (Ruminant) Furthermore, samples with a $\Delta^{13}C$ value less than $-1.26$‰ (2 s.d. from the wild ruminant mean) and a SRR% less than 64% (upper quartile of ruminant adipose and below the lower quartile of aquatic resources) are also assigned to this source.

(6) (Animal) A further generic animal content is inferred by the presence of cholesterol oxidation or biohydrogenation by-products, occasionally associated with cholesterol[71,72].

(7) (Plant) Plant epicuticular waxes are inferred by the presence of long chain $n$-alkanes (>$C_{20}$) with a clear odd to even carbon chain number prevalence[73].

(8) (Plant) Plant epicuticular waxes are also composed of long chain (>$C_{20}$) saturated fatty acids (LCSFA) with an even to odd carbon chain number prevalence[73]. Because small amounts of long chain fatty acids can also be present in most animal tissues[74], only samples with >15% LCSFA (LCSFA/saturated fatty acids) are assigned to this source.

(9) (Plant) Use of the palmitic to stearic fatty acid ratio (P:S) to infer pottery content is highly criticized[71]. Because fatty acid distribution is prone to modification by alteration processes, a direct comparison between modern and archaeological fats is not possible. Nevertheless, because shorter chain fatty acids are more labile and disappear preferentially, the P:S ratio will not increase artificially due to the degradation process. Plant products generally show a high predominance of palmitic acid compared with animal fat. Consequently, it is likely that samples with a high P:S ratio contained plants. We used a P:S ratio threshold of 4, as proposed by Dunne et al.[75].

(10) (Plant) Similarly a $C_{12}$:$C_{14}$ ratio has been proposed as a criterion to differentiate plant and animal fats[76] and is also unlikely to increase due to degradation. A conservative threshold of 1 was used to assign a plant source.

(11) (Plant) α-Amyrin, β-amyrin and their amyrone derivative are used as a plant proxy. They are common terpenoids among angiosperms but are also sometimes found in sediments. Nevertheless, a recent study has demonstrated that when those compounds are found, sometimes in high abundance, they are likely to be endogenous and are derived from plant processing, notably from *Viburnum* berries known to be frequently found in these pots[24].

(12) (Plant) Another criteria used to identify plant lipids is the presence of phytosterol and derivatives (stigmasterol, campesterol and so on).

(13) (Plant) Various cereals, fruits and non-leafy plants have a relative high abundance of the APAA-$C_{18}$ E isomer compared to the H isomer, that are unlikely to result from either mixing or extensive heat alteration[66]. We assigned cases to this category when the APAA-$C_{18}$ E:H ratio was higher than 4.

(14) (Plant) 2-Hydroxy fatty acids derived from animal or plant sphingolipids. Long chain 2-hydroxy fatty acids are notably quite abundant in the extract of *Viburnum* berries. We used their presence as a tentative criteria for plant lipids.

(15) (Heating) We have also defined a series of criteria to infer the heating of the commodities. The presence of APAAs implies that unsaturated fatty acids have been subjected to heating (at least 1 h at >200 °C), easily achieved through boiling or roasting the vessel contents in an open fire[64–66].

(16) (Heating) Similarly, long chain ketones (16-hentriacontanone, 16-tritriacontanone and 18-pentatriacontanone) are a by-product of pro-acted heating of fatty acids and triglycerides[77,78].

(17) (Heating) Finally, benzene polycarboxilic acids are a by-product of condensed charred organic matter or 'black carbon' formed during the acid-catalysed extraction procedure[79].

Different classes of product were assigned to each sample according to those criteria: aquatic resources (1 AND 2, OR 3), ruminant fats (4 OR 5), non-specific animals (6 NOT 2–5), plant resources (OR 7–14) and heating (OR 15–17).

## Biomolecular distance

Using the 17 traits described above, a binary presence–absence matrix was composed, indicating which samples contain biomarkers that signal the presence of fatty acids derived from ruminants, non-ruminant terrestrial animals, aquatic resources, plants and heating. R scripts aggregated these to the analytical level of each vessel, then each site, and computed distance matrices using the Jaccard coefficient, as per the ceramic data.

## Landscape analysis

Storing a database of site locations in a geographic information system (GIS), we generated distance matrices containing the pairwise geodesic great-circle distance between each pair of sites using the haversine formula. To investigate whether landscape heterogeneity impacts the strength of cultural connections, which straight-line distance would be blind to, the GIS was also used to find the length of the least-cost path connecting each pair of sites. This measurement was derived from analysis of a 100-m digital elevation model of Eurasia obtained from the ASTER Global DEM v.3 (ref. [80]), using the r.cost and r.drain algorithms in GRASS GIS[81]. Because least-cost paths generate a single solution, they are sensitive to relatively minor obstacles, which is potentially a problem for the low-lying steppe regions. To redress this, we applied Circuitscape analysis[82], in which the landscape is modelled using electrical resistance rather than mechanical cost, and calculated the difficulty in moving from site to site considering all possible paths. These results were stored in a distance matrix. The Julia package circuitscape[83] was used to undertake this analysis.

## Spatial–temporal modelling

Guided by the earliest dated material from hunter-gatherer ceramic contexts immediately east of our study region[84], the site of Mergen 6 in western Siberia, dating to ~6,500 cal BC, was used to apply a temporal gradient to models of the spread of ceramic traditions west of the Urals. The least-cost distance from this to each dated site was used in a reduced major axis linear regression model against time to calculate the diffusion rate for the adoption of pottery by hunter-gatherers. The posterior probability distribution of the start of pottery use at each site was modelled in OxCal v.4.4 using Markov Chain Monte-Carlo inference[85], with samples drawn from this process used in multiple permutations to express a confidence interval for the regression arising from chronological uncertainty. To generate a model, the length of the least-cost path between nodes of a regular grid of points and a rasterized surface was interpolated from this using thin plate spline regression in SAGA GIS[86]. Next, raster algebra in GRASS GIS was used to parametrize each pixel using the results of the spatio-temporal regression and the date of the point of origin. Contour lines (isochrones) were drawn using the r.contour module in GRASS GIS. We repeated this process for different sites in western Siberia and Central Asia, but the results did not alter significantly.

## Correspondence analysis

The contingency tables were subjected to correspondence analysis using two-factor principal canonical correlation, and the corresponding row scores plotted to visualize the structure of the presence–absence data. The R package MASS[87] together with custom scripts contained in the Supplementary Information were used to undertake this analysis.

## Mantel correlation tests

First, we calculated the Jaccard distance between sites using the pairwise ratio of traits present at two sites and the number of traits in total (the ratio of intersection over union, subtracted from 1). To compare the geographical, spatio-temporal, ceramic and biomolecular distance matrices, we calculated the Pearson product–moment correlation coefficient between each pair of distance matrices using the Mantel test. As well as providing a correlation coefficient that expressed the strength of the correlation between each dataset, this procedure used 500 bootstrap resamples to test a null hypothesis that there was no relation between each pair of matrices. Mantel correlograms were calculated using 213 km distance classes, with significantly positive or negative correlations identified using a permutation test. The R package ecodist[88] was used to undertake this analysis.

## Phylogenetic network analysis

We used the neighbour-joining network construction algorithm neighborNet[89] to create phylogenetic networks of the trait data, using a subset of the data limited to sites for which we had some control of chronology. This agglomerative, exploratory method constructs a 'splits graph' with each node (site) neighbouring nodes with similar traits. Each node is modelled as having a unique evolutionary history, with the network representing a composite of these histories, the connections representing the evolutionary distances between nodes. The R package phangorn[90] was used to undertake this analysis.

*Editorial Note: S. Telizhenko and V. Manko requested removal from the author list in response to Russia's invasion of Ukraine.*

## Reporting summary

Further information on research design is available in the Nature Portfolio Reporting Summary linked to this article.

## Data availability

Data files including all the ceramic data and contingency tables for the organic residue traits are contained in an electronic repository accessed via the following URL: https://doi.org/10.5281/zenodo.6619101.

## Code availability

Scripts in the R language for reproducing the analysis are available in an electronic repository accessed via the following URL: https://doi.org/10.5281/zenodo.6619101.

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

## Acknowledgements

This project has received funding from the European Research Council (ERC) under the European Union's Horizon 2020 research and innovation programme (grant agreement number 695539, The Innovation, Dispersal and Use of Ceramics in NW Eurasia) to C.H. Research at the site of Dąbki was conducted under the National Science Centre, Poland (grant agreement number 2017/27/B/HS3/00478). H.K.R. acknowledges the British Academy for funding. E.O.'s work was supported by the Estonian Research Council (grant agreement numbers PXX MOBERC14 and PSG492). The funders had no role in study design, data collection and analysis, decision to publish or preparation of the manuscript. We would like to thank Przemysław Muzolf and Błażej Muzolf for access to Lutomiersk-Wrząca ceramics. The project team expresses sincere gratitude to all colleagues and institutions across the study region who have collaborated on, and supported, the research programme since it began in 2016. Two co-authors, S. Telizhenko and V. Manko, contributed materials to this research, but requested that their names be removed from the author list. We thank them for their contributions. The project team dedicates this paper to the memory of V. Lozovski who warmly encouraged and supported research into the ceramic technology of hunter-gatherers across the study region.

## Author contributions

O.E.C., C.H. and J.M. conceived and designed the study. E.D., E.O., K. Adamczak, K. Andreev, V.A., M.C., A.C.-Z., I.E., T.G., A.G., T.M.G., D.H., M.I., J.K., V.K., N.K., E.K., A.K., S.K., O.L., A.M., N.N., G.P., G.S., A.

Skorobogatov, R.V.S., A. Surkov, O.T., M.T., A.T., V.T., A.A.V., A.W. and A.I.Y. provided materials and information on archaeological context. E.D. undertook the analysis of ceramics. A.L., M.B., B.C., E.O., H.K.R. and H.T. undertook laboratory analyses. J.M. analysed radiocarbon data. T.R.M. developed the modelling approach. A.L. and T.R.M. developed computer code for modelling and data analysis, and analysed the data. A.L., O.E.C. and T.R.M. led the writing of the paper with input from E.D., H.K.R., C.H., J.M. and H.P.

## Competing interests

The authors declare no competing interests.

## Additional information

**Correspondence and requests for materials** should be addressed to T. Rowan McLaughlin.

[1]State Hermitage Museum, Saint Petersburg, Russia. [2]British Museum, London, UK. [3]BioArCh, Department of Archaeology, University of York, York, UK. [4]University of Tartu, Tartu, Estonia. [5]Institute of Pre- and Protohistory, Kiel, Germany. [6]Institute of Archaeology, Nicolaus Copernicus University, Toruń, Poland. [7]Samara State University of Social Sciences and Education, Samara, Russia. [8]Institute of History of the National Academy of Sciences of Belarus, Minsk, Belarus. [9]Institute of Archaeology and Ethnology Polish Academy of Sciences, Poznań, Poland. [10]State Autonomous Institution for Heritage Research and Production, Astrakhan, Russia. [11]National History Museum of Latvia, Rīga, Latvia. [12]Russian Research Institute for Cultural and Natural Heritage, Saint Petersburg, Russia. [13]Institute of Archaeology, National Academy of Sciences of Ukraine, Kyiv, Ukraine. [14]East Onega Archaeological Expedition, Vologda, Russia. [15]Institute of Language, History and Literature, Komi Scientific Center of Ural Branch of RAS, Syktyvkar, Russia. [16]Cherepovets State University, Cherepovets, Russia. [17]Ivanovo State University, Ivanovo, Russia. [18]Institute for the History of Material Culture RAS, Saint Petersburg, Russia. [19]The Vologda State Museum, Vologda, Russia. [20]Lithuanian Institute of History, Vilnius, Lithuania. [21]Voronezh Archaeological Society, Voronezh, Russia. [22]Lipetsk State Pedagogical University PP Semenov-Tyan-Shan, Lipetsk, Russia. [23]Archaeological Society of Kuban, Rostov-on-Don, Russia. [24]Don Archaeological Society, Rostov-on-Don, Russia. [25]Podlachian Museum in Białystok, Białystok, Poland. [26]Research Center for the Preservation of Cultural Heritage, Saratov, Russia. [27]Centre for Baltic and Scandinavian Archaeology, Schleswig, Germany. [28]Present address: Maynooth University, Maynooth, Ireland. [29]Prague, Czech Republic. [30]These authors contributed equally: Ekaterina Dolbunova, Alexandre Lucquin, T. Rowan McLaughlin. ✉e-mail: rowan.mclaughlin@mu.ie

# Reporting Summary

## Statistics

For all statistical analyses, confirm that the following items are present in the figure legend, table legend, main text, or Methods section.

| n/a | Confirmed | |
|---|---|---|
| ☐ | ☒ | The exact sample size (*n*) for each experimental group/condition, given as a discrete number and unit of measurement |
| ☐ | ☒ | A statement on whether measurements were taken from distinct samples or whether the same sample was measured repeatedly |
| ☐ | ☒ | The statistical test(s) used AND whether they are one- or two-sided *Only common tests should be described solely by name; describe more complex techniques in the Methods section.* |
| ☐ | ☒ | A description of all covariates tested |
| ☐ | ☒ | A description of any assumptions or corrections, such as tests of normality and adjustment for multiple comparisons |
| ☐ | ☒ | A full description of the statistical parameters including central tendency (e.g. means) or other basic estimates (e.g. regression coefficient) AND variation (e.g. standard deviation) or associated estimates of uncertainty (e.g. confidence intervals) |
| ☐ | ☒ | For null hypothesis testing, the test statistic (e.g. *F*, *t*, *r*) with confidence intervals, effect sizes, degrees of freedom and *P* value noted *Give P values as exact values whenever suitable.* |
| ☐ | ☒ | For Bayesian analysis, information on the choice of priors and Markov chain Monte Carlo settings |
| ☐ | ☒ | For hierarchical and complex designs, identification of the appropriate level for tests and full reporting of outcomes |
| ☐ | ☒ | Estimates of effect sizes (e.g. Cohen's *d*, Pearson's *r*), indicating how they were calculated |

*Our web collection on statistics for biologists contains articles on many of the points above.*

## Software and code

Policy information about availability of computer code

| Data collection | MSD Chemstation version F.01.03.2357; Mass Hunter version B.07.01; NIST MS search version 2.2; Isodat version 3.0; IonOS version 4; Google Sheets |
|---|---|
| Data analysis | R version 4.0 (with packages vegan 2.5.7, ecodist 2.0.7, lmodel2 1.7.2, dplyr 1.0, scatterpie 0.18, and their dependencies); custom scripts (available at doi.org/10.5281/zenodo.6619101); GRASS GIS version 7.4; SAGA GIS version 7; Julia version 1.6; circuitscape version 5; IsoMemo version 1.9; OxCal version 4.4 |

For manuscripts utilizing custom algorithms or software that are central to the research but not yet described in published literature, software must be made available to editors and reviewers. We strongly encourage code deposition in a community repository (e.g. GitHub). See the Nature Portfolio guidelines for submitting code & software for further information.

## Data

Policy information about availability of data

All manuscripts must include a data availability statement. This statement should provide the following information, where applicable:

- Accession codes, unique identifiers, or web links for publicly available datasets
- A description of any restrictions on data availability
- For clinical datasets or third party data, please ensure that the statement adheres to our policy

Data files including all the ceramic data and contingency tables for the organic residue traits are contained in an electronic repository accessed via the following URL: doi.org/10.5281/zenodo.6619101

## Human research participants

Policy information about studies involving human research participants and Sex and Gender in Research.

| | |
|---|---|
| Reporting on sex and gender | N/A |
| Population characteristics | N/A |
| Recruitment | N/A |
| Ethics oversight | NA |

Note that full information on the approval of the study protocol must also be provided in the manuscript.

# Field-specific reporting

Please select the one below that is the best fit for your research. If you are not sure, read the appropriate sections before making your selection.

☒ Life sciences          ☐ Behavioural & social sciences          ☐ Ecological, evolutionary & environmental sciences

For a reference copy of the document with all sections, see nature.com/documents/nr-reporting-summary-flat.pdf

# Life sciences study design

All studies must disclose on these points even when the disclosure is negative.

| | |
|---|---|
| Sample size | Sample size was determined by the quantity of archaeological remains curated in museum collections for the cultural groups under analysis. |
| Data exclusions | No data were excluded from initial analyses; samples that failed to yield lipid concentrations below a predetermined analytical minimum of 5 µg per g of potsherds and 100 µg per g for charred surface deposits were excluded from trait-based analyses. |
| Replication | Samples for mass spectrometry were not measured in duplicate but replicated measurements of a lab standard were run routinely with each batch of samples. |
| Randomization | Samples were classified by geographical region considering the location of the archaeological sites. |
| Blinding | Blinding was not relevant for our study; some of the samples were selected by their morphological and decorative features. |

# Reporting for specific materials, systems and methods

We require information from authors about some types of materials, experimental systems and methods used in many studies. Here, indicate whether each material, system or method listed is relevant to your study. If you are not sure if a list item applies to your research, read the appropriate section before selecting a response.

## Materials & experimental systems

| n/a | Involved in the study |
|---|---|
| ☒ | ☐ Antibodies |
| ☒ | ☐ Eukaryotic cell lines |
| ☐ | ☒ Palaeontology and archaeology |
| ☒ | ☐ Animals and other organisms |
| ☒ | ☐ Clinical data |
| ☒ | ☐ Dual use research of concern |

## Methods

| n/a | Involved in the study |
|---|---|
| ☒ | ☐ ChIP-seq |
| ☒ | ☐ Flow cytometry |
| ☒ | ☐ MRI-based neuroimaging |

## Palaeontology and Archaeology

Specimen provenance | Information on specimen provenance, including site location, is detailed in the paper, including contextual details for new radiocarbon dates.

Specimen deposition | Pottery samples were destroyed during the organic residue analyses

Dating methods | New radiocarbon dates were obtained using terrestrial samples (herbivore bones, wood charcoal, plant macrofossils) found in association with the samples in this study, and used to develop chronological models for a select number of sites using OxCal 4.4 and the IntCal20 calibration dataset.

☒ Tick this box to confirm that the raw and calibrated dates are available in the paper or in Supplementary Information.

Ethics oversight | No new archaeological excavations were performed so ethical approval was not required. Permission for destructive analysis was given by the institutions curating the material.

Note that full information on the approval of the study protocol must also be provided in the manuscript.

