## [Peer Review File · Nature Human Behaviour]

Peer Review Information

Journal: Nature Human Behaviour

Manuscript Title: The transmission of pottery technology amongst prehistoric European hunter-gatherers

Corresponding author name(s): T. Rowan McLaughlin

Editorial Notes:

Redactions – unpublished data	Parts of this Peer Review File have been redacted as indicated to maintain the confidentiality of unpublished data.
Redactions – confidential patient information	Parts of this Peer Review File have been redacted as indicated to maintain patient confidentiality.
Redactions – published data	Parts of this Peer Review File have been redacted as indicated to remove third-party material.

Reviewer Comments & Decisions:

Decision Letter, initial version:
--

9th May 2022

Dear Dr McLaughlin,

Thank you once again for your manuscript, entitled "The transmission of pottery technology amongst prehistoric European hunter-gatherers," and for your patience during the peer review process.

Your manuscript has now been evaluated by 3 reviewers, whose comments are included at the end of this letter. Although the reviewers find your work to be of interest, they also raise some important concerns. We are very interested in the possibility of publishing your study in Nature Human Behaviour, but would like to consider your response to these concerns in the form of a revised manuscript before we make a decision on publication.

To guide the scope of the revisions, the editors discuss the referee reports in detail within the team, including with the chief editor, with a view to (1) identifying key priorities that should be addressed in

revision and (2) overruling referee requests that are deemed beyond the scope of the current study.

In the case of your manuscript, as you will see from the Reviewer comments below, Reviewer 2 is particularly concerned about the fact that a solvent-based extraction method was not used. We ask that you perform these complementary analyses or clearly explain why they were not possible to perform. Please do not hesitate to get in touch if you would like to discuss this further.

Your revised manuscript must comply fully with our editorial policies and formatting requirements. Failure to do so will result in your manuscript being returned to you, which will delay its consideration. To assist you in this process, I have attached a checklist that lists all of our requirements. If you have any questions about any of our policies or formatting, please don't hesitate to contact me.

In sum, we invite you to revise your manuscript taking into account all reviewer and editor comments. We are committed to providing a fair and constructive peer-review process. Do not hesitate to contact us if there are specific requests from the reviewers that you believe are technically impossible or unlikely to yield a meaningful outcome.

We hope to receive your revised manuscript within three months. I would be grateful if you could contact us as soon as possible if you foresee difficulties with meeting this target resubmission date.

- Include a "Response to the editors and reviewers" document detailing, point-by-point, how you addressed each editor and referee comment. If no action was taken to address a point, you must provide a compelling argument. When formatting this document, please respond to each reviewer comment individually, including the full text of the reviewer comment verbatim followed by your response to the individual point. This response will be used by the editors to evaluate your revision and sent back to the reviewers along with the revised manuscript.
- Highlight all changes made to your manuscript or provide us with a version that tracks changes.

[REDACTED]

We look forward to seeing the revised manuscript and thank you for the opportunity to review your work. Please do not hesitate to contact me if you have any questions or would like to discuss these revisions further.

Sincerely,

Charlotte Payne

Charlotte Payne, PhD

Senior Editor
Nature Human Behaviour

Reviewer expertise:

Reviewer #1: european hunter-gatherers; population modeling

Reviewer #2: residue analysis; radiocarbon dating

Reviewer #3: residue analysis; food pottery

REVIEWER COMMENTS:

Reviewer #1:

Remarks to the Author:

This paper on the spread of pottery use in west Eurasia hunter-gatherer groups will be of broad interest to archaeologists and to other scholars with interests in cultural evolution. It is based on a new data set bringing together information on ceramic residues and a wide range of other information, all from the same set of vessels. The data choices are clearly explained and justified and the same is true of the methods used. I find the results convincing and the conclusions drawn from them likewise. Some of the general discussion at the end is a bit unfocussed and brings in concepts such as 'archaeological cultures' that have not previously been introduced.

'These ceramics are unlikely part of the initial dispersal of ceramics and are excluded from statistical analysis of their stylistic and technological traits.' Explain why.

Fig 5 caption '). Correlation coefficients where $p < 0.05$ are not shown. ' Shoul be $>$ not $<$

'Pottery subsequently spread rapidly westwards towards the Baltic, covering over 3000 km in three to four centuries, a process too rapid to be explained by demic diffusion' Could have tested for the existence of a demographic wave.

'It is an open question how far archaeological 'cultures' can reflect discrete groups of people, wider communication networks, or are in some cases merely the product of discontinuous sampling from continuous variation. Here, it seems the latter is more conceptually applicable, but that innovation occurred more slowly than adoption through communication networks.' We haven't heard anything about how the patterns in the data relate to 'cultures'. Not clear what is the basis for the claim that here we are dealing with discontinuous sampling from continuous variation.

'The low occurrence of pottery vessels from individual sites does not impact the analytical approaches used which depend on the number of traits shared between sites rather than the number not shared.' But the denominator is based on the total traits present at the two sites

Reviewer #2:

Remarks to the Author:

This manuscript reports new results from a study concerned with the timing, adoption and diffusion of Middle Holocene hunter gatherer pottery in the Western Russian and the Eastern European regions. Pottery played a critical role in the culture-social and economic development of H-G communities across Eurasia, and remains an important tool and source of information for researchers.

The cultural, economic and environmental factors driving the uptake of prehistoric pottery in Northern Eurasia has been debated intensely among scholars of late, with plenty of publications addressing this question through chemical analysis of ancient food residues.

The present manuscript goes beyond this, however, and intersects - using different statistical approaches - a large number of chemically analyzed pottery samples with more traditional archaeological, radiometric and geographic evidence. The findings presented support a relatively quick spread of HG pottery. Moreover, these results indicate that pottery adoption and use were transmitted through cultural interactions across super-regional communication networks.

These kinds of syntheses that zoom out and look at pottery function at an inter-regional level can reveal important insights into prehistoric subsistence and cultural interactions, capturing the attention of - not only archaeologists and academics, but also the general public.

The present study is well-thought and executed, and supported by a large group of experts in Eastern European archaeology and archaeological sciences. The methods employed and study results are fairly well communicated to the reader, and the conclusions drawn are sensible. The manuscript reads well and is properly structured, although the Supplemental Information should be revised for typos.

Having said that, I have a few questions about the methodology and would like the authors to address the questions and comments below.

Pending these responses, I am recommending the publication of this paper, and I believe it is a good fit for this journal.

L146: an acidified methanol extraction was used according to Courel et al. Did you also test some samples, as was done in Courel et al., using a solvent-based extraction method (for instance, using a DCM/Methanol mixture) and see whether a solvent extraction would have yielded identical or at least comparable results?

L147: What was the threshold amount, exactly?

L166: If you only did the acidified methanol extraction, and considering how this approach tends to reduce the detectability of plant-related lipids, I'm not surprised that lipids associated with plants were recorded only in trace quantities. It seems that by omitting the solvent-based extraction step you have potentially introduced a bias that understates the proportion of plant lipids in the pots analyzed. You should mention this possibility in the text.

Also, if plant lipids detected were so frequently (74%) associated with animal products, a question

emerges whether some or most of these plant remains are derived from the burial environment (e.g. roots) rather than the ancient use of the vessel. Did you find any evidence that would support an archaeological source for the plant lipids?

L168-169: The mentioned absence of plant resins and tars, which should be more prevalent in the dataset, could, again, be explained by your methodological choices, not necessarily HG container function.

L170-173: Did you find conclusive evidence, i.e. detected the key compounds relating to this complex chemical mixture, for the presence of beeswax in an acid-extracted ceramic sample? Acidified methanol treatment is by no means optimal for the detection of beeswax, and rather one should carry out a solvent-based extraction (see Roffet-Salque et al.) when searching for signs of beeswax in ancient samples. If you didn't employ a solvent-based extraction, it's no wonder that beeswax was not visible in the sample set. This should be mentioned in the text, also especially since you compare these results to a study (reference no. 27) that recorded a widespread use of beeswax by employing a solvent-based extraction method. Also, wouldn't it make sense to mention the site where this beeswax sample originated from?

L195: should read "this is most clearly seen"

L206-208: To me this statement seems to somewhat contradict with one of the key findings of this paper, i.e. that the detected regional variation in pottery use is derived from diversified culinary practices, which is high in spite of similar environmental settings and resource availability across the study region.

L390: Did you find vicinal dihydroxy fatty acids in any of the samples, and if so, why wasn't this biomarker among the 17 interpretative criteria? Vicinal diols are known to derive from marine/aquatic organisms and their presence should help settle the source of lipids in some of the samples, thus making the model more accurate.

Reviewer #3:

Remarks to the Author:

The spread of pottery in hunter-gather communities would reflect the culture interaction among different groups. North and east Europe is an ideal region for understanding the mechanism of pottery adoption and use. Although some previous research discussed the pottery function over time, when and where the pottery were introduced and the function of pottery in north and east Europe still need more work. This study integrated carbon dating data, organic residue analysis and typology analysis for pottery from the wide region in north and east Europe, demonstrating how the pottery technology spread from the East to the West. This work will promote further understanding the culture evolution during this area. Here, I have a few concerns.

1) For the samples without aquatic biomarkers, many points locate in the unique range of freshwater animals, what's the possible animals? Also, some points in the $\Delta^{13}C$ range of ruminant animal fat with much C4 input, is it possible that the wild ruminant animals, such as deer, ate many C4 plants at that time in the north or east Europe?

2) it's better to compare the spread rate and spatial-temporal routes of agriculture through demic diffusion, also the westward spread rate of Yamnaya culture about 3300 BC.

Author Rebuttal to Initial comments

Reviewer #1:

Remarks to the Author:

This paper on the spread of pottery use in west Eurasia hunter-gatherer groups will be of broad interest to archaeologists and to other scholars with interests in cultural evolution. It is based on a new data set bringing together information on ceramic residues and a wide range of other information, all from the same set of vessels. The data choices are clearly explained and justified and the same is true of the methods used. I find the results convincing and the conclusions drawn from them likewise. Some of the general discussion at the end is a bit unfocussed and brings in concepts such as 'archaeological cultures' that have not previously been introduced.

We thank the reviewer very much for their kind comments and constructive criticism. We respond to the point about archaeological cultures below.

'These ceramics are unlikely part of the initial dispersal of ceramics and are excluded from statistical analysis of their stylistic and technological traits.' Explain why.

We have now explained that they are excluded because "they are the product of later phenomena and influences from multiple sources, including agricultural societies" Have also added a reference to a paper by Guminski (2020) in support of this.

Fig 5 caption '). Correlation coefficients where $p < 0.05$ are not shown. ' Shoul be $>$ not $<$

Fixed, many thanks.

'Pottery subsequently spread rapidly westwards towards the Baltic, covering over 3000 km in three to four centuries, a process too rapid to be explained by demic diffusion' Could have tested for the existence of a demographic wave.

We agree, and indeed in our original submission we considered this. However the comparison was made deep in the supplementary information. We have now edited the main text of the paper to expand on this point as follows, and added several references in support:

Notably, this is several times faster than the spread of Neolithic pottery from the Middle East into the Mediterranean and western Europe (Jordan et al. 2016; Bocquet-Appel et al. 2012; Isern et al. 2017). Through forward modelling, it has been shown that demic diffusion can drive the spread of ancient technology in such cases where the rate of spread is much less than what we have determined for hunter-gatherer pottery in Europe (Fort and Méndez 1999; Fort 2012)

'It is an open question how far archaeological 'cultures' can reflect discrete groups of people, wider communication networks, or are in some cases merely the product of discontinuous sampling from continuous variation. Here, it seems the latter is more conceptually applicable, but that innovation occurred more slowly than adoption through communication networks.' We haven't heard anything about how the patterns in the data relate to 'cultures'. Not clear what is the basis for the claim that here we are dealing with discontinuous sampling from continuous variation.

Our thanks to the reviewer for pointing out this oversight. The idea of archaeological culture is actually implicit throughout our paper, although we have taken a somewhat unconventional approach to its analysis. In the opening part of our paper we introduced the idea that pottery decoration and form, etc. 'represent human knowledge fossilized in the artefact' and our paper would 'reconstruct connections between societies, separated by geographic distance or time, using a set of bio-statistical tools to evaluate the relatedness of archaeological cultures based on traits'. To underscore the message that we are referring to technological traits as 'culture', we have modified the introduction as follows:

These attributes, which are sometimes taken together and interpreted as 'archaeological cultures' represent human knowledge fossilized in the artefact. They can be used to reconstruct connections between societies ...

Furthermore, in the discussion section, we have now restated that groupings we detect in pottery morphology and decoration are "sometimes identified as archaeological 'cultures'" and we have added a reference to a key Shennan *et al.* paper (already cited in the introduction) that contains a

fulsome discussion of the issues of archaeological ‘discontinuous sampling’ with respect to pottery traditions.

'The low occurrence of pottery vessels from individual sites does not impact the analytical approaches used which depend on the number of traits shared between sites rather than the number not shared.' But the denominator is based on the total traits present at the two sites

Apologies that our point here was confused; we were merely introducing the rationale for using statistical methods like bootstrapping in the first place. We have modified the text to read:

Because there are low occurrences of pottery vessels from some individual sites, our analytical approaches employ resampling procedures to test a null hypothesis that patterns in the traits shared between sites occur randomly.

Reviewer #2:

Remarks to the Author:

This manuscript reports new results from a study concerned with the timing, adoption and diffusion of Middle Holocene hunter gatherer pottery in the Western Russian and the Eastern European regions. Pottery played a critical role in the culture-social and economic development of H-G communities across Eurasia, and remains an important tool and source of information for researchers.

The cultural, economic and environmental factors driving the uptake of prehistoric pottery in Northern Eurasia has been debated intensely among scholars of late, with plenty of publications addressing this question through chemical analysis of ancient food residues.

The present manuscript goes beyond this, however, and intersects - using different statistical approaches - a large number of chemically analyzed pottery samples with more traditional archaeological, radiometric and geographic evidence. The findings presented support a relatively quick spread of HG pottery. Moreover, these results indicate that pottery adoption and use were transmitted through cultural interactions across super-regional communication networks.

These kinds of syntheses that zoom out and look at pottery function at an inter-regional level can reveal important insights into prehistoric subsistence and cultural interactions, capturing the attention of - not only archaeologists and academics, but also the general public.

The present study is well-thought and executed, and supported by a large group of experts in Eastern European archaeology and archaeological sciences. The methods employed and study results are fairly well communicated to the reader, and the conclusions drawn are sensible.

The manuscript reads well and is properly structured, although the Supplemental Information should be revised for typos.

Having said that, I have a few questions about the methodology and would like the authors to address the questions and comments below.

Pending these responses, I am recommending the publication of this paper, and I believe it is a good fit for this journal.

We thank the reviewer very much for their positivity and extremely helpful feedback.

L146: an acidified methanol extraction was used according to Courel et al. Did you also test some samples, as was done in Courel et al., using a solvent-based extraction method (for instance, using a DCM/Methanol mixture) and see whether a solvent extraction would have yielded identical or at least comparable results?

As noted above to the editor, we undertook a small pilot study to evaluate the different extraction methods within the constraints of the study. In total, we undertook solvent extraction and acid extraction on different portions of 100 samples. The results showed conclusively that the acid extraction yielded greater quantities of lipids (the average yield was 8.9 times higher using acidified methanol extraction) and a higher number of samples provided interpretable data.

In terms of data quality between the two extraction procedures, (Correa-Ascencio and Evershed, 2014) noted that the loss of triacylglycerols (TAGs) and wax esters are the main drawbacks of acidified methanol extraction compared to solvent, as the ester bond will break during extraction.

For the pilot, we selected samples that were lipid rich or showing a profile suggesting the presence of waxes, plant product or tar as well as others to serve as a “control”. We recovered a TAGs from nine samples (with a further six containing traces of TAGs), with a distribution corresponding to the presence of ruminant products (n=7). In contrast, we were able to systematically detect ruminant fats on a much greater range of samples by applying a

combination of molecular and isotopic criteria to the methanol acid extracts. So the presence of TAGs would not have significantly enhanced our interpretation.

In the 100 solvent extracts, only one sample contained wax esters that can be attributed to beeswax and was already reported in 2020 (Courel *et al. Royal Society Open Science*). Plant wax esters were recovered in two samples but these were also attributed to plants using the criteria based on acid methanol extraction. We also confirmed the identification of birch bark tar in three samples but failed to detect any further evidence of this product. Again, the analysis of the solvent extract did not enhance our interpretation of the subset analyzed.

For more clarity we have added the following sentence:

In addition, a total of 100 samples were also solvent extracted following established procedures (Courel et al., 2020) to investigate the presence and distribution of triacylglycerols (TAGs) or wax esters, but these failed to generate additional information.

We have also now included the solvent extract result on TAGs and wax esters in the online repository of data and code associated with this paper, with a new version available via the following DOI:

<https://doi.org/10.5281/zenodo.6619101>

L147: What was the threshold amount, exactly?

We added the thresholds to the text:

(i.e. $>5 \mu\text{g g}^{-1}$ for potsherds and $>100 \mu\text{g g}^{-1}$ for charred surface deposits)

L166: If you only did the acidified methanol extraction, and considering how this approach tends to reduce the detectability of plant-related lipids, I'm not surprised that lipids associated with plants were recorded only in trace quantities. It seems that by omitting the solvent-based extraction step you have potentially introduced a bias that understates the proportion of plant lipids in the pots analyzed. You should mention this possibility in the.

As noted by (Correa-Ascencio and Evershed, 2014) whilst the the acid methanol fails to specifically identify plant products based on their wax ester distribution, it does nonetheless

provide a way of indicating the presence of such molecules through their degradation products that are amenable to this extraction procedure. If we had identified widespread markers for degraded waxes throughout the assemblages, we would have had more convincing grounds for undertaking more detailed solvent extraction.

Other usual plant biomarkers (i.e. alkanes, phytosterol, terpenes...) are recovered during the acid methanol extraction procedure. Nevertheless we used conservative criteria that probably underestimates the presence of some commodities.

We have added the following sentence:

This count has to be considered as a minimal conservative number of occurrences of a resource as the absence of a criterion is not always related to the absence of a resource.

Also, if plant lipids detected were so frequently (74%) associated with animal products, a question emerges whether some or most of these plant remains are derived from the burial environment (e.g. roots) rather than the ancient use of the vessel. Did you find any evidence that would support an archaeological source for the plant lipids?

Removal of the external surface of the sherd was done before sampling in order to reduce the possible contamination due to the burial environment. We do not observe a correlation between burial environment/site and lipid composition as it might be expected if those compounds were derived from the sedimentary matrix.

We also observed direct evidence, using scanning electron microscopy (SEM), of fragments of plant tissues preserved within a selection of charred deposits preserved on the pottery vessel surface. This will provide a stand alone study which is forthcoming.

We edited the following sentence for clarity:

Plant products are frequent (587/1425), sometimes with fragments of carbonized plant tissues visible within the charred deposit (Bondetti et al., 2020; Courel et al., 2021), but were probably not the main commodities. Typical clear leafy plant lipid profiles are rare and plant biomarkers are

generally identified in only **small or** trace quantities and in 74% of their instances they are associated with aquatic or terrestrial animal fats.

L168-169: The mentioned absence of plant resins and tars, which should be more prevalent in the dataset, could, again, be explained by your methodological choices, not necessarily HG container function.

We respectfully disagree with the reviewer on this point. Acid extraction doesn't preclude the detection of resin and tar, even if a solvent extraction is needed to resolve all the constituents of these substances. Diterpenes from coniferous resin (i.e. abietic acid, dehydroabietic acid, 7-oxodehydroabietic acid) are released and methylated during acidified methanol extraction so these will be detected using our approach. Triterpenes such as betulin or lupeol and various derivatives are also extracted and identifiable even without silylation. As stated, these compounds were mostly absent or detected in trace amounts incompatible with a “lipid profile typical of plant resins and tars (i.e. where di- and triterpenes dominate)”.

For clarity, we edited the following sentence and updated the information in the SI database:

There was an almost complete absence **(29/1425)** of lipid profiles typical of plant resins and tars (i.e. where di- **or** triterpenes **are prominent in the extract**),

L170-173: Did you find conclusive evidence, i.e. detected the key compounds relating to this complex chemical mixture, for the presence of beeswax in an acid-extracted ceramic sample? Acidified methanol treatment is by no means optimal for the detection of beeswax, and rather one should carry out a solvent-based extraction (see Roffet-Salque et al.) when searching for signs of beeswax in ancient samples. If you didn't employ a solvent-based extraction, it's no wonder that beeswax was not visible in the sample set. This should be mentioned in the text, also especially since you compare these results to a study (reference no. 27) that recorded a widespread use of beeswax by employing a solvent-based extraction method. Also, wouldn't it make sense to mention the site where this beeswax sample originated from?

We agree with the reviewer that solvent extraction is a better choice for the unequivocal detection of beeswax. Nevertheless, an acidified methanol is perfectly suitable for identifying

samples potentially containing beeswax through the distribution of fatty acids and alkanes hydrolysed from any intact wax esters (Correa-Ascencio and Evershed, 2014). In fact, these compounds are more likely to survive and be released from ceramics using this method than intact wax esters through solvent extraction. Such a profile was detected only in a single sample from Rosenhof in Northern Germany. In this case, the presence of beeswax was then confirmed by solvent extraction and was reported in Courel *et al* (2020). We have not reported this again here but we are confident that beeswax was not present in the wider range of samples included here otherwise we would have identified a greater proportion of samples with evidence of degraded waxes.

We edited the following sentence to add the site and the reference:

Similarly, only one sample found at Grube-Rosenhof LA 58 (Courel *et al.*, 2020) contained beeswax contrasting with their higher prevalence in Early Neolithic agricultural pottery (Roffet-Salque *et al.*, 2015).

L195: should read "this is most clearly seen"

Fixed, thanks

L206-208: To me this statement seems to somewhat contradict with one of the key findings of this paper, i.e. that the detected regional variation in pottery use is derived from diversified culinary practices, which is high in spite of similar environmental settings and resource availability across the study region.

The reviewer makes an interesting point; the variation is indeed high. However, this statement is intended simply to point out that the statistical differences between the cultural traits are greater than the statistical differences between pottery use. We have changed the statement to read:

Thus despite regional variation, the use of the pots was nonetheless more consistent over the study region than the cultural factors that influenced the way they were made.

L390: Did you find vicinal dihydroxy fatty acids in any of the samples, and if so, why wasn't this biomarker among the 17 interpretative criteria? Vicinal diols are known to derive from

marine/aquatic organisms and their presence should help settle the source of lipids in some of the samples, thus making the model more accurate.

Vicinal dihydroxy fatty acids are more easily observed in extracts after derivatisation by trimethylsilylation. This derivatisation process was not undertaken systematically in the dataset that consists of new and previously published data, hindering the systematic record of those compounds and therefore we have not used this as a criterion. Instead, we used the alkylphenyl fatty acids and %SRR ratio as corroborative biomarkers for aquatic foods. Nevertheless, the number of samples with aquatic biomarkers should always be seen as a minimum number. We clarified this with the following sentence:

This count has to be considered as a minimal conservative number of occurrences of a resource as the absence of a criteria is not always related to the absence of a resource.

Reviewer #3:

Remarks to the Author:

The spread of pottery in hunter-gatherer communities would reflect the culture interaction among different groups. North and east Europe is an ideal region for understanding the mechanism of pottery adoption and use. Although some previous research discussed the pottery function over time, when and where the pottery were introduced and the function of pottery in north and east Europe still need more work. This study integrated carbon dating data, organic residue analysis and typology analysis for pottery from the wide region in north and east Europe, demonstrating how the pottery technology spread from the East to the West. This work will promote further understanding the culture evolution during this area.

Here, I have a few concerns.

1) For the samples without aquatic biomarkers, many points locate in the unique range of freshwater animals, what's the possible animals?

It is difficult to attribute a clear origin to the samples plotting within the “freshwater” range but without aquatic biomarkers. A simple explanation could be that the aquatic biomarkers may not be preserved in those samples. However, the “freshwater” range overlaps with isotope values of fatty acids obtained from a variety of C3 plants, including wild plants such as acorn (Lucquin et al., 2016).

We did not add a specific “plant” range. Firstly, the isotopic values of local reference material is not developed enough considering the variety of plant products that may have been consumed by prehistoric hunter gatherers. Secondly, we are not using these ellipses (Fig 3a and 3b) to infer pottery use, merely to highlight the range of values obtained and show that the distribution is wider in the samples with aquatic biomarkers than those without, as would be expected.

Also, some points in the $\Delta^{13}\text{C}$ range of ruminant animal fat with much C4 input, is it possible that the wild ruminant animals, such as deer, ate many C4 plants at that time in the north or east Europe?

Most wild ruminants in the study area have a C3 diet. Nevertheless, C4 plant input to animal diet is possible in some regions (i.e. Northern Caspian, Low Volga) and Reindeer also have more enriched fatty acid values than other Cervidae species. However, the criteria used to indicate the presence of ruminant fat in the pottery vessels (i.e. the $\Delta^{13}\text{C}$ value and the SRR%) are independent of animal diet so this would not affect our interpretation.

Nevertheless, we have updated Fig 3 and now included the fatty acid isotopic values from a wider range of modern species (Reindeer and Saiga antelope) to better reflect the anticipated range. Again, please note that we are not using Fig 3a and 3b for interpretative purposes but rather we are highlighting the spread of data.

2) it's better to compare the spread rate and spatial-temporal routes of agriculture through demic diffusion, also the westward spread rate of Yamnaya culture about 3300 BC.

This point about demic diffusion was also raised by reviewer 1; please see our response above.

As for the comparison with Yamnaya culture, we believe the cultural and technological differences (metals, domesticated horses, etc) introduce too many confounding variables for any comparison to be useful at present. Furthermore, we are not aware of any study that has directly measured the Yamnaya spread using archaeological radiocarbon data (inference of the rapidity of the Yamnaya spread derives from genomic studies e.g. (Olalde et al., 2018)). Whilst it is an interesting point that our study constitutes an early example of rapid movement of culture within the same region later identified as an origin of the Yamnaya-like ancestry, we feel a detailed comparison is beyond the scope of our paper and best left to future work.

We hope the editors find these revisions satisfactory. In the text we have highlighted all changes in yellow and in red type. We have also re-formatted our Supplementary Information in-line with journal guidelines, and updated our cross-references in the paper. Many thanks for considering our work for publication.

Rowan McLaughlin

8th June 2022

References cited in this response:

- Bondetti, M., Scott, S., Lucquin, A., Meadows, J., Lozovskaya, O., Dolbunova, E., Jordan, P., Craig, O.E., 2020. Fruits, fish and the introduction of pottery in the Eastern European plain: Lipid residue analysis of ceramic vessels from Zamostje 2. *Quat. Int.* 541, 104–114.
- Correa-Ascencio, M., Evershed, R.P., 2014. High throughput screening of organic residues in archaeological potsherds using direct acidified methanol extraction. *Anal. Methods* 6, 1330–1340.
- Courel, B., Meadows, J., Carretero, L.G., Lucquin, A., McLaughlin, R., Bondetti, M., Andreev, K., Skorobogatov, A., Smolyaninov, R., Surkov, A., Vybornov, A.A., Dolbunova, E., Heron, C.P., Craig, O.E., 2021. The use of early pottery by hunter-gatherers of the Eastern European forest-steppe. *Quat. Sci. Rev.* 269, 107143.
- Courel, B., Robson, H.K., Lucquin, A., Dolbunova, E., Oras, E., Adamczak, K., Andersen, S.H., Astrup, P.M., Charniauski, M., Czekaj-Zastawny, A., Ezepenko, I., Hartz, S., Kabaciński, J., Kotula, A., Kukawka, S., Loze, I., Mazurkevich, A., Piezonka, H., Piličiauskas, G., Sørensen, S.A., Talbot, H.M., Tkachou, A., Tkachova, M., Wawrusiewicz, A., Meadows, J., Heron, C.P., Craig, O.E., 2020. Organic residue analysis shows sub-regional patterns in the use of pottery by Northern European hunter-gatherers. *R Soc Open Sci* 7, 192016.
- Lucquin, A., Gibbs, K., Uchiyama, J., Saul, H., Ajimoto, M., Eley, Y., Radini, A., Heron, C.P., Shoda, S., Nishida, Y., Lundy, J., Jordan, P., Isaksson, S., Craig, O.E., 2016. Ancient lipids document continuity in the use of early hunter-gatherer pottery through 9,000 years of Japanese prehistory. *Proc. Natl. Acad. Sci. U. S. A.* 113, 3991–3996.
- Olalde, I., et al , 2018. The Beaker phenomenon and the genomic transformation of northwest Europe. *Nature* 555, 190.
- Roffet-Salque, M., et al 2015. Widespread exploitation of the honeybee by early Neolithic farmers. *Nature* 527, 226–230.

Decision Letter, first revision:

30th June 2022

Dear Dr. McLaughlin,

Thank you for submitting your revised manuscript "The transmission of pottery technology amongst prehistoric European hunter-gatherers" (NATHUMBEHAV-22030506A). It has now been seen by the original referees and their comments are below. As you can see, the reviewers find that the paper has improved in revision. We will therefore be happy in principle to publish it in Nature Human Behaviour, pending minor revisions to comply with our editorial and formatting guidelines.

We are now performing detailed checks on your paper and will send you a checklist detailing our editorial and formatting requirements within a week. Please do not upload the final materials and make any revisions until you receive this additional information from us.

Sincerely,

Charlotte Payne

Charlotte Payne, PhD
Senior Editor
Nature Human Behaviour

Reviewer #1 (Remarks to the Author):

The reviewers have addressed all my queries to my satisfaction and I am happy to recommend acceptance.

Reviewer #2 (Remarks to the Author):

Thank you for these comments. I am happy with the corrections you've made based on my feedback.

Final Decision Letter:

Dear Dr McLaughlin,

We are pleased to inform you that your Article "The transmission of pottery technology amongst prehistoric European hunter-gatherers", has now been accepted for publication in Nature Human Behaviour.

Please note that *Nature Human Behaviour* is a Transformative Journal (TJ). Authors whose manuscript was submitted on or after January 1st, 2021, may publish their research with us through the traditional subscription access route or make their paper immediately open access through payment of an article-processing charge (APC). Authors will not be required to make a final decision about access to their article until it has been accepted. IMPORTANT NOTE: Articles submitted before January 1st, 2021, are not eligible for Open Access publication. Find out more about Transformative Journals

With best regards,

Charlotte Payne

Charlotte Payne, PhD
Senior Editor
Nature Human Behaviour